# The Multifaceted Gene 275 Embedded in the PKS-PTS Gene Cluster Was Involved in the Regulation of Arthrobotrisin Biosynthesis, TCA Cycle, and Septa Formation in Nematode-Trapping Fungus *Arthrobotrys oligospora*

**DOI:** 10.3390/jof8121261

**Published:** 2022-11-29

**Authors:** Jiao Zhou, Qun-Fu Wu, Shu-Hong Li, Jun-Xian Yan, Li Wu, Qian-Yi Cheng, Zhi-Qiang He, Xu-Tong Yue, Ke-Qin Zhang, Long-Long Zhang, Xue-Mei Niu

**Affiliations:** State Key Laboratory for Conservation and Utilization of Bio-Resources & Key Laboratory for Microbial Resources of the Ministry of Education, School of life Sciences, Yunnan University, Kunming 650091, China

**Keywords:** *Arthrobotrys oligospora*, *AOL_s00215g275*, sesquiterpenyl epoxy-cyclohexenoids (SECs), polyketide synthase-prenyltransferase (PKS-PTS) hybrid pathway, multifunction, retinoic acid induced-1 (Rai1) protein, cortactin and (Vit1) proteins, TCA cycle, mitochondria

## Abstract

The predominant nematode-trapping fungus *Arthrobotrys oligospora* harbors a unique polyketide synthase-prenyltransferase (PKS-PTS) gene cluster *AOL_s00215g* responsible for the biosynthesis of sesquiterpenyl epoxy-cyclohexenoids (SECs) that are involved in the regulation of fungal growth, adhesive trap formation, antibacterial activity, and soil colonization. However, the function of one rare gene (*AOL_s00215g275* (*275*)) embedded in the cluster has remained cryptic. Here, we constructed two mutants with the disruption of *275* and the overexpression of *275*, respectively, and compared their fungal growth, morphology, resistance to chemical stress, nematicidal activity, transcriptomic and metabolic profiles, and infrastructures, together with binding affinity analysis. Both mutants displayed distinct differences in their TCA cycles, SEC biosynthesis, and endocytosis, combined with abnormal mitochondria, vacuoles, septa formation, and decreased nematicidal activity. Our results suggest that gene *275* might function as a separator and as an integrated gene with multiple potential functions related to three distinct genes encoding the retinoic acid induced-1, cortactin, and vacuolar iron transporter 1 proteins in this nematode-trapping fungus. Our unexpected findings provide insight into the intriguing organization and functions of a rare non-biosynthetic gene in a biosynthetic gene cluster.

## 1. Introduction

It is not unusual for there to be ‘other’ incongruous genes embedded in a biosynthetic gene cluster that is responsible for the biosynthesis of secondary metabolites. Previous studies have suggested that such ‘other’ genes appear to be unnecessary for the formation of secondary metabolites but display important roles in fungal self-protection mechanisms, including duplication of the metabolite target, detoxification of the metabolites, and efflux of the metabolites [1]. However, limited attention has been paid to the functions of such genes in new biosynthetic gene clusters.

*Arthrobotrys oligospora* is a typical nematode-trapping fungus that is widespread in diverse terrestrial and aquatic environments and is an ideal natural agent against the detrimental crop parasites known as nematodes [2,3,4]. Previous studies of *A. oligospora* have identified a number of unique polyketide–terpenoid hybrid metabolites—including oligosporons, anthrobotrisins, and arthrosporols—involved in the regulation of mycelial growth and fusion, conidial development, adhesive trap formation, antibacterial and antiviral activities, and soil colonization [5,6,7,8]. This class of unique metabolites, which consist of a farnesyl-unit-derived sesquiterpenyl unit linked to a 6-methylsalicylic acid (6-MSA)-derived epoxy polyketide nucleus and belong to the sesquiterpenyl epoxy-cyclohexenoids (SECs), are widely distributed in fungi and include the well-known potent antifungal yanuthones from *Aspergillus* sp. and *Penicillium* sp. and the notorious phytotoxic macrophorins from the fungus *Macrophoma*, which causes fruit rot in apples [9,10,11]. Among these metabolites, anthrobotrisins A-D, have been regarded as characteristic chemotaxonomic markers for *A. oligospora* and could be used as chemical indicators for nematode-associated or infected environments [12,13].

A gene cluster *AOL_s00215g* (*215g*) containing 10 genes (Figure 1A)—including a 6-MSA polyketide synthase (PKS) gene *283* and a prenyltransferase (PTS) gene *276*—is responsible for this class of hybrid metabolites [14,15,16]. Combination of the construction of the mutants with the deletion of each gene in the *215g* gene cluster, along with comparison of the metabolic profiles between the mutants and WT, revealed that genes *273*−*274* and *276*−*283* were involved in the biosynthesis of the hybrid metabolites, except for one gene (*275*) [15]. Moreover, the homologue of gene *275* was only missing in the *Yan* and *Mac* gene clusters that are responsible for the biosynthesis of SECs in *Aspergillus* sp. and *Penicillium* sp. (Figure 1A). Thus, two questions arise: why the gene *275* is embedded in the *215g* cluster of the nematode-trapping fungus, and what kind of role the gene *275* plays in the fungus.

In the present study, we found that no conserved domain could be predicted in 275 using routine threshold values for bioinformatics analysis. The protein 275 only shared a limited identity to the retinoic acid induced-1 (Rai1) protein, src substrate cortactin, and vacuolar iron transporter 1 (Vit1) protein (Figure 1B). Next, we constructed two mutants: one mutant Δ*275* with disruption of the gene *275* in *A. oligospora*, and the other mutant OE-*275* with the replacement of a strong promoter *PtrpC* upstream of the gene *275*. Then, we compared the morphology, growth, trap formation, nematicidal activity, resistance to chemical stressors, and transcriptomic and metabolic profiles of the mutants and wild type (WT). The results revealed that the differentially expressed genes (DEGs) were highly enriched in the TCA cycle and diminished in membrane transport and ABC transporters. Interestingly, all of the genes in the *215g* cluster responsible for the anthrobotrisin biosynthesis were dramatically upregulated in Δ*275* vs. WT. Importantly, the genes involved in period circadian and actin-related proteins were the most downregulated. Metabolic profiles revealed that the TCA cycle and the hybrid metabolites were most dramatically regulated in the mutants. Moreover, Δ*275* displayed swollen hyphae, while OE-*275* had abnormal septa. A series of detailed analyses revealed that the Δ*275* mutants displayed phenotypic and infrastructural traits due to the lack of three putative functions related to Rai1, cortactin, and Vit1. Our results suggest that *275* can perform multiple functions in fungal growth and development via regulating the SEC biosynthesis, TCA cycle, and septa formation.

## 2. Materials and Methods

### 2.1. Fungal Strain and Culture Conditions

*A. oligospora* YMF1.3170 was obtained from the State Key Laboratory for Conservation and Utilization of Bio-Resources and Key Laboratory for Microbial Resources of the Ministry of Education. PDA (potato 200 g/L, glucose 10 g/L, agar 15 g/L), TG (10 g/L tryptone, 10 g/L glucose, 15 g/L agar), TYGA (10 g/L tryptone, 5 g/L yeast extract, 10 g/L glucose, 5 g/L molasses, 15 g/L agar), and YMA (5 g/L yeast extract, 10 g/L malt extract, and 15 g/L agar) media were used for analyzing mycelial growth and related phenotypic traits. For the flask culture, the strain was cultured on PDA medium at 28 °C for 10 days to obtain conidia, and then the conidia were inoculated in 250 mL of liquid YPG medium (5 g of yeast extract, 5 g of peptone, and 20 g of glucose per liter) with a final concentration of 1 × 10^5^ conidia/mL at 28 °C at 180 rpm.

### 2.2. Sequence and Phylogenetic Analysis of 275 in A. oligospora

Predicted *275* in *A. oligospora* was annotated by BLAST searches against protein databases and InterProScan searches against protein domain databases. The amino acid sequence of *275* in *A. oligospora* was downloaded from GenBank, as were the sequences from different fungi. The amino acid sequences from different fungi were analyzed using the DNAman software package, version 5.2.2 (Lynnon Biosoft, Vaudreuil, QC, Canada). A neighbor-joining tree was constructed using the Mega 5.1 software package (PSU, USA). The web-based analysis platform antiSMASH 2.0 (https://fungismash.secondarymetabolites.org/; accessed on 16 April 22) was applied to perform genome mining of the biosynthetic gene clusters. The functional domains were predicted using the default parameters of InterProScan.

### 2.3. Construction of Mutant Δ275 and OE-275

A modified homologous recombination method was applied to construct the vectors for the mutant Δ*275* and OE-*275*. All primers used in this study are provided in Appendix A. The DNA fragments (5′ flanks and 3′ flanks) were purified using the Ribo EXTRACT Universal DNA Purification Kit (Tsingke, Beijing, China). The purified 5′ flank DNA fragment was inserted into the specific enzyme-digested PUC19-1300-D-HYB vector via the In-Fusion method to produce the PUC19-1300-D-HYB-5′ vector. Then, the 3′ flank DNA fragment was inserted into the specific enzyme-digested PUC19-1300-D-HYB-5′ vector to generate the PUC19-1300-D-HYB-5′–3′ vector. The homologous fragment was amplified and purified as described previously. The gene *275* was inserted into a PUC19 plasmid containing the strong promoter *PtrpC* and terminator *TtrpC* from *Aspergillus nesterus*. The plasmid was amplified as described previously. The purified homologous fragments cut from the plasmid were transformed into *A. oligospora* protoplasts. Transformation colonies were selected after incubation at 28 °C for 2–4 days, and every single colony was transferred to a new plate containing PDA. After incubation at 28 °C for 5 days, genomic DNA of putative transformants was extracted and verified by PCR to check for integration of the target in the genome. The mutant deficient in the target gene was screened out and confirmed by PCR and sequencing analysis.

### 2.4. Quantitative Real-Time PCR Analysis

The WT strain and OE-275 mutant were cultured in PDA medium at 28 °C for 6 days. The vegetative hyphae were harvested in 9-cm Petri dishes, which were then incubated at 28 °C for 3 and 5 d. After induction, hyphae were collected and frozen immediately in liquid nitrogen. Total RNA was extracted from all samples with the AxyPrep Multisource RNA Miniprep Kit (Axygen, Jiangsu, China). The extracted RNA was then reverse transcribed to cDNA with the FastQuant RT Kit with gDNase (Takara, Kusatsu, Japan). The cDNA was used as the template for a qRT-PCR assay, which was conducted to analyze the transcription levels of candidate genes. The gene-specific qRT-PCR primers were designed with Primer3 software, and the β-tubulin gene (Tub, *AOL_s00076g640*) was used as an internal standard. The qRT-PCR analysis was performed as previously described [16]. The relative transcription level (RTL) of each gene was calculated as the ratio of the transcription level in the deletion mutant to the transcription level in the WT strain at a given timepoint according to the 2^−ΔΔCt^ method [16].

### 2.5. Ultrahigh-Performance Liquid Chromatography–Mass Spectrometry (UPLC−MS) Analysis

The WT and mutant strains were inoculated in flasks containing 250 mL of potato dextrose broth (PDB) and cultured at 28 °C (180 rpm) for 7 days. The fermentation broth was extracted with an equal volume of ethyl acetate, and the organic layer was evaporated and dissolved in 1 mL of menthol and filtered through a 0.22 μm filter membrane. All samples were then analyzed using a UPLC system coupled with a Q Exactive Focus Orbitrap mass spectrometer (Thermo Fisher, Waltham, MA, USA) and equipped with an Agilent Zorbax ODS 4.6 × 250 mm^2^ column (Agilent, Santa Clara, CA, USA) using the electrospray ionization (ESI) mode. The metabolites extracted from the WT and mutant strains of *A. oligospora* were analyzed by UPLC−MS according to the methods described in the literature [15].

### 2.6. Fungal Colony Growth and Morphology

The WT and mutant strains were cultured on PDA, TYGA, and YMA media at 28 °C, and their growth rates and colony morphology were observed. Hyphal morphology was examined using 5-day-old PDA cultures grown at 28 °C. The fungal strains were stained with calcofluor-white (CFW, Sigma-Aldrich, St. Louis, MO, USA)—a fluorescent dye that naturally binds to cellulose and chitin. The hyphae were collected using a sterilized coverslip. The fungal strains were stained with 20 μg/mL CFW and then observed under an inverted fluorescence microscope (Nikon, Tokyo, Japan).

### 2.7. Fungal Conidial Production and Germination

The conidial yields of the fungal strains were determined as previously described [15,16]. Three replicates for each strain were initiated by spreading a 100 μL aliquot of a 10^5^ conidia mL^−1^ suspension per CMY plate, followed by 10 days of incubation at 28 °C. Three colony plugs (5 mm in diameter) were taken from each plate culture, and the conidia on each plug were washed off into 1 mL of 0.02% Tween 80 through 10 min of vibration. The conidial concentration in the suspension was determined by microscopic counting in a hemocytometer and converted to the number of conidia per cm^2^ of plate culture as an estimate of conidial yield. Moreover, conidial formation was observed with an established side-shot approach. To calculate the conidial germination rates, 50 μL aliquots of 10^5^ conidia mL^−1^ suspensions of the WT strain and mutants were used to inoculate WA medium. The samples were incubated at 28 °C, and the germinated conidia were counted at 4 h intervals until 24 h post-inoculation. The phenotypic analyses were repeated three times.

### 2.8. Multi-Stress Assays

Chemical stressors including sorbitol, Congo red, sodium dodecyl sulfate (SDS), and H_2_O_2_ were used for the evaluation of fungal stress responses. The colonies of the WT and mutant strains were initiated with 9 cm diameter hyphal disks at 28 °C for 6 days on plates of TG or supplemented with each of the following chemical stressors: NaCl (0.1 mol/L, 0.2 mol/L, 0.3 mol/L) and sorbitol (0.25, 0.5, 0.75 mM) for osmotic stress; Congo red (0.1, 0.2, 0.3 mg/mL) and sodium dodecyl sulfate (SDS) (0.01, 0.02, 0.03%) for cell-wall-disrupting agents; and H_2_O_2_ (5, 10, 15 mmol/L) for oxidative stress. The diameter of each colony was measured and then calculated with GraphPad 9.3.0 (San Diego, CA, USA). The experiments were performed with at least three replicates.

### 2.9. Trap Formation and Nematicidal Activity Assays

WT and mutant strains were cultured on 9 cm agar plates at 28 °C for 3–4 days, and then about 400 nematodes (*Caenorhabditis elegans*) were introduced to the cultures. After 12 h, traps and captured nematodes per plate were observed under a microscope and counted at specific timepoints.

### 2.10. Endocytosis Assay

To evaluate endocytosis, hyphae of the WT and Δ*275* strains were cultured on PDA plates for 4–5 days. The sterilized coverslips were inserted into the PDA medium at a slant (45° angle), and aerial hyphae were cultured on the coverslip for 5 days. The coverslip with aerial mycelia was incubated with FM4-64 staining solution (SynaptoRed C2, Biotium, Fremont, CA, USA; 10 μL of FM4-64 was diluted to a final concentration of 4 μM in 50 mM Tyrode solution) and immediately placed under a fluorescence microscope (Nikon, Tokyo, Japan) for observation [17].

### 2.11. Transmission Electron Microscopy (TEM)

For TEM analysis, ultrathin sections of hyphal cells were prepared and examined as previously described [16]. The samples of the transmission electron microscope (TEM) were pretreated with 4% paraformaldehyde and then incubated with 2.5% glutaraldehyde (Sigma) in phosphate buffer (pH = 7.4) overnight at 4 °C and post-fixed in 1% OsO_4_. The samples were dehydrated in a gradient ethanol series, embedded in Spurr resin, and stained with 2% uranyl acetate and Reynold’s lead solution. The samples were examined under an H–7650 transmission electron microscope (Hitachi, Tokyo, Japan).

### 2.12. Quantitative Analysis of Arthrobotrisins and TCA Metabolites and Amino Acids

Twenty standard amino acids and seven standard compounds of the tricarboxylic acid (TCA) cycle were dissolved in methanol at concentrations of 10 μM. The relative contents of metabolites of TCA and amino acids were calculated by evaluating the peak areas according to standard substances. The metabolites were identified through retention time indices and mass spectra of peaks. Compounds were designated as target metabolites if they were identified with a match of 950 on a scale of 0–1000 with a retention index deviation of 3.0. The semi-quantitative analysis of the main compounds was carried out by internal normalization with the area of each compound.

### 2.13. Transcriptomic Analysis

The WT and mutant strains were cultured in PDB medium (potato 200 g/L, glucose 10 g/L) at 28 °C for 7 days. Sequencing of mycelial samples was performed by Shanghai Majorbio Bio-pharm Technology Co., Ltd. (Shanghai, China), and the data were analyzed using the Majorbio Cloud Platform (www.majorbio.com (accessed on 10 March 2022)). The transcripts per kilobase million method (TPM) was used to calculate the expression level of each transcript to identify differentially expressed genes (DEGs). Functional enrichment analysis identified the DEGs that were significantly enriched in the GO and KEGG terms (*p* ≤ 0.05) through GO and KEGG analysis.

### 2.14. Protein Modeling and Docking

Protein structural modeling of 275 was performed using AlphaFold v2.0 with ColabFold v1.0 (https://deepmind.com/research/open-source; accessed on 24 July 2022) [18,19]. Minimized energy optimization of structures of the precursors, intermediates, and their derivatives in the biosynthetic pathway for anthrobotrinsins in *A*. *oligospora* was processed using ChemBio3D Draw (version 20.0.0.41, Cambridge Soft, Cambridge, MA, USA). The processing of the 275 protein (hydrogenation, and Gasteiger charge for merging nonpolar hydrogen atoms) was performed using MGLTools (v1.5.6) in AutoDock Vina software (https://autodock.scripps.edu/download-autodock4/; accessed on 24 August 22) [19]. The original pdb file format was converted to the pdbqt file format, which can be recognized by the AutoDock Vina program. Furthermore, the docking active center (including all residues around the original ligand) was set through the grid box function in the software, based on the size of the root-mean-square deviation (RMSD) of the docked ligand molecule and the original ligand molecule with reasonable docking parameter settings. RMSD ≤ 4 was set for the threshold for conformation of the ligand after docking to match the conformation of the original ligand. The best docking mode was selected as the lowest obtained binding energy result.

### 2.15. Statistical Analysis

All experiments were repeated at least three times. Data are presented as the mean ± standard deviation (SD). GraphPad Prism v9.0.2 software (San Diego, CA, USA) was used for statistical analysis. Comparisons were performed using the two-tailed Student’s *t*-test. Probabilities of *p* < 0.05 were used as the threshold for determining significant differences (* *p* < 0.05; ** *p* < 0.01; *** *p* < 0.001).

## 3. Results and discussion

### 3.1. Sequence and Phylogenetic Analyses of 275

The gene *275* consists of only one intron and encodes a protein of *275* amino acids with an isoelectric point of 5.3 and a molecular mass of 30.7 kDa. Surprisingly, no conserved domains were identified in *275* using InterProScan and the Conserved Domain Database (CDD). Importantly, only one unknown protein was retrieved from fungi when searching against UniProt (Parameter: E-Threshold < 0.1) with *275*. The protein from *Arthrobotrys flagrans* displayed a high similarity to *275* (76% identity). Other retrieved sequences were mainly from mammals, fish, insects, and flies. All of the proteins displayed a low similarity to *275*, ranging from 23% to 29%. To illustrate the homology of 13 retrieved sequences with-known protein functions to the *275* protein, a maximum-likelihood (ML) phylogenetic tree under the best-fit amino acid substitution model with 200 bootstrap replications was constructed (Figure 1B). The E value was adjusted to 1 in order to include those protein sequences with higher similarity to *275*. The phylogenetic tree revealed that 275 fell into a clade containing retinoic acid induced-1 (Rai1) protein, src substrate cortactin, and vacuolar iron transporter 1 (Vit1) protein.

Analysis of the subcellular localization of the *275* protein with Cell-PLoc indicated that 275 might be located in the nucleus (12), mitochondria (10), cytoplasm (4), and cytoskeleton (1). Interestingly, Rai1 was localized to the nucleus, cortactin was localized to the cytoplasm, and Vit1 was localized to the plasma membrane. Moreover, we found that the Rai1 protein covers the first half of 275, while cortactin covers most of the latter half of 275 (Figure 1B), and Vit1 covers almost 80% of the 275 protein. Previous studies suggest that Rai1 is involved in the circadian regulation of gene expression [20,21]. Cortactin acts as an actin-binding protein by binding and activating Arp2/3, playing critical roles in the regulation of the cytoskeleton, cell migration, and endocytosis [22,23,24]. Vit1 in the plant *Eucalyptus grandis* is a H^+^-coupled antiporter for metal ions, which provides the vacuolar H^+^-ATPase with an important function in metal storage [25,26].

### 3.2. Function of 275 in Regulating Fungal Growth and Morphology

The mutants Δ*275* with the disruption of 275 and OE-*275* with the replacement of the strong promoter *PtrpC* were constructed and identified via PCR analyses (Figure 2A,B). qRT-PCR analysis revealed that the transcription level of *275* in OE-*275* was 2.68 times and 3.97 times higher than that in the WT at 3 and 5 days, respectively, suggesting that OE-*275* displayed a significantly increased transcription level of *275* compared with the wild-type strain (WT) (Figure 2C). Three media—including one nutrient-deficient medium (PDA, which is usually used for trap formation) and two nutrient-rich media (YMA and TYGA)—were evaluated for fungal colony growth and morphology. Compared with the WT, both Δ*275* and OE-*275* displayed significantly decreased fungal growth on the nutrient-rich media (YMA and TYGA), while no significant difference in fungal growth was observed on the nutrient-deficient medium (PDA) (Figure 2D). Interestingly, at day 6 on all three media, both Δ*275* and OE-*275* displayed distinct fungal colonies with different morphology from that of WT (Figure 2D). Obviously, the WT displayed much denser and thicker aerial mycelia than both mutants. Moreover, we found that both Δ*275* and OE-*275* displayed distinct mycelia zones compared to the WT (Figure 2D). These results suggest that the disruption of gene *275* or the enhanced transcription level of *275* could lead to significantly varied fungal colony growth, aerial mycelia, and mycelia zones.

### 3.3. Roles of 275 in Conidial Production, Germination, Trap Formation, and Nematicidal Activity

The mutants Δ*275* and OE-*275* displayed quite opposite conidial production and conidial germination compared with the WT at 28 °C (Figure 3A). The mutant Δ*275* showed a 204.4% increase in conidial formation and a 7.6% increased germination rate, while OE-*275* displayed significantly decreases in conidial formation and germination rate (by 33.1% and 11.3%, respectively) (Figure 3A,B). Similarly, Δ*275* and OE-*275* also displayed quite opposite trap formations compared with the WT. The mutant Δ*275* showed slightly decreased trap formation, while OE-*275* displayed significantly enhanced trap formation (by 98.4% and 65.0%, respectively), compared with the WT at 24 h and 36 h (Figure 3C,D). The mutant Δ*275* exhibited significantly decreased nematicidal activity (by 18.5%) compared with the WT at 24 h (Figure 3D,E). Surprisingly, the mutant OE-*275* also exhibited significantly decreased nematicidal activity (by 29.6%) compared with the WT at 24 h (Figure 3E). All of the results suggested that the gene *275* was involved in inhibiting conidial formation, germination, and nematicidal activity but promoting trap production. We found that the decreased trap formation in Δ*275* was in stark contrast to the dramatically increased trap formation in the other mutants that were deficient in each biosynthetic gene in the gene cluster *215g* [15].

### 3.4. The Association of 275 with Multiple Stress Responses

Surprisingly, we found that the mutant Δ*275* showed significantly increased resistances to several chemical stressors—including two cell-wall-perturbing agents (SDS and Congo red), two osmotic agents (NaCl and sorbitol), and the oxidant H_2_O_2_—on the TYGA medium (Figure 4A,B). Δ*275* displayed increased colony growth when exposed to Congo red, NaCl, and sorbitol at all of the tested concentrations. Moreover, Δ*275* also exhibited distinct aerial mycelia on 15 mM H_2_O_2_ compared to the WT. Notably, OE-275 also displayed distinct aerial morphology on SDS and enhanced growth on NaCl, sorbitol, and 10–15 mM H_2_O_2_ (Figure 4A,B). Interestingly, we also found that the inner-ring mycelia of OE-*275* became much thinner than those of the WT on Congo red, NaCl, and H_2_O_2_ (Figure 4A). These results suggest that the disruption of *275* or the enhanced transcription of 275 could exert intensive influences on the fungal resistances and significantly regulate the fungal morphology and responses to all of the tested chemical stressors.

### 3.5. Transcriptional Link of 275 to the PKS-PTS Hybrid Pathway, TCA Cycle, Membrane Transport, and Redox Homeostasis

Transcriptomic analysis displayed that there were significant differences between Δ*275* and *OE-275* and the WT (Figure 5A). There were 227 genes expressed only in the WT, 136 genes only in Δ*275*, and 114 genes only in *OE-275*. A total of 9502 genes were co-expressed between the WT, Δ*275*, and *OE-275* (Figure 5B). Compared with the WT, 943 and 902 genes were differentially expressed in Δ*275* and *OE-275*, respectively. Among them, 447 genes were significantly upregulated and 496 genes were significantly downregulated in Δ*275* (Figure 5C). In *OE-275*, 441 genes were significantly upregulated and 461 genes were significantly downregulated (Figure 5D).

Interestingly, the gene *276*, which is immediately adjacent to *275* and encodes a PTS for biosynthesis of the first hybrid precursor, ranked the first among the upregulated genes in Δ*275* vs. the WT due to having the lowest Padj value. Moreover, among the top 20 upregulated genes in Δ*275* vs. the WT, the gene *276* also ranked the first due to having by far the highest FPKM value (19.6 in the WT vs. 422.7 in Δ*275*) (Figure 5E, Appendix A). Further analysis of the transcriptional levels of all of the genes involved in the PKS-PTS hybrid pathway (273–283) indicated that all of the genes were dramatically upregulated in Δ*275*. Among them, two genes—*279* and *277*, which were both involved in the formation of the key epoxy ring in the first PKS-PTS hybrid precursor—displayed similar increases in their transcriptional levels to that of gene 276 (Figure 5E).

Among the top 20 downregulated genes, *79g256*, which encodes the period circadian protein, displayed by far the highest FPKM value (1101.44 in the WT vs. 29.41 in Δ*275*), with a 36.4-fold increase (Figure 5F, Appendix A). Moreover, three genes—including *76g531*, encoding a putative L-fucose-proton symporter; *4g471*, encoding neuromodulin; and *6g248*, encoding actin-related protein 2/3 complex subunit 5 (Arp2/3)—were also in the top five FPKM values among these downregulated genes in Δ*275* vs. the WT (Figure 5F). Previous studies suggest that neuromodulin is among the most prominent substrates of protein kinase C (PKC) in the brain and that phosphorylation of neuromodulin by PKC functions to allow maximal Ca^2+^/CaM binding and vesicle release in mammals [27,28]. Interestingly, previous studies indicate that Arp2/3 is a protein that can be recruited by cortactin to exist in actin microfilaments, facilitating and stabilizing nucleation sites for actin branching, thereby playing an important role in promoting cell migration and endocytosis [25,26].

KEGG enrichment analysis of all of the upregulated genes in Δ*275* vs. the WT indicated that the TCA cycle was the most upregulated in Δ*275* (Figure 5G), while arginine biosynthesis, pyruvate metabolism, nitrogen metabolism, and alanine/aspartate/glutamate metabolism ranked the 2nd to the 5th, respectively. KEGG enrichment analysis of all of the downregulated genes in Δ*275* vs. the WT exhibited that membrane transport and ABC transporters were the most downregulated in Δ*275* (Figure 5H). KEGG analysis of the differentially expressed genes in OE-*275* vs. the WT (Appendix A) indicated that the most upregulated pathways included glycolysis/gluconeogenesis, phenylalanine/tyrosine/tryptophan biosynthesis, pyruvate metabolism, alanine/aspartate/glutamate metabolism, and the TCA cycle (Figure 5I), while the most downregulated pathways were membrane transport, ABC transporters, fatty acid degradation, glycosylphospatidylinositol (GPI)-anchored proteins, and SNARE interactions in vesicular transport (Figure 5J). A previous study suggested that, in mammalian cells, GPI-anchored proteins are concentrated in lipid rafts that are involved in receptor-mediated signal transduction pathways and membrane trafficking [29].

Further GO analysis of analysis of all of the upregulated genes in Δ*275* vs. the WT indicated that oxidoreductase, amide transmembrane transporter, and nucleoside monophosphate kinase activities were the most upregulated in Δ*275* (Figure 6A). GO analysis of all of the downregulated genes in Δ*275* vs. the WT showed that ABC-type transporter activity was the most downregulated in Δ*275* (Figure 6B). GO analysis of the differentially expressed genes in OE-*275* vs. the WT showed that the amide transmembrane transporter was the most upregulated (Figure 6C) while cysteine-type endopeptidase activity was the most downregulated in OE-*275* (Figure 6D). A previous study suggested that cysteine-type endopeptidase activity was mainly involved in antioxidant activity [30]. Vacuolar processing enzyme (VPE) is a cysteine-type endopeptidase that mediates programmed cell death (PCD) by provoking vacuolar rupture and initiating the proteolytic cascade leading to PCD [30].

### 3.6. 275 Regulated the Production of Unique PKS-PTS Hybrid Metabolites, TCA Metabolites, and Amino Acids

UPLC–MS analysis of the metabolic profiles of Δ*275*, OE-*275*, and the WT was performed. Both Δ*275* and OE-*275* displayed quite similar metabolic profiles to the WT, except that Δ*275* had two more peaks at 13 and 28 min (Figure 7A). Detailed analysis of the hybrid arthrobotrisins showed that Δ*275* had 2.1-fold higher arthrosporol levels than the WT, and there was no significant difference in OE-*275* (Figure 7B). Furthermore, citrate dramatically increased by 911.4% in Δ*275* vs. the WT, while malate decreased by 62.9%. Small significant increases were observed for pyruvate (35.1%) and succinate (7.6%) in Δ*275* vs. the WT (Figure 7B). Surprisingly, cis-aconitate, D-isocitrate, and α-ketoglutarate could be not detected in Δ*275*. Furthermore, all of the above TCA metabolites—except citrate—could not be detected in OE-*275* either (Figure 7C). However, OE-275 displayed quite similar composition of aromatic acids to the WT, except that OE-275 had 438.0% more serine, 67.2% more alanine, and 215.9% more arginine but 70.8% less glutamine and 34.7% less glutamic acid than the WT (Figure 7D). The mutant Δ*275* displayed large increases in serine (401.2%), histine (103.5%), alanine (204.6%), and arginine (1127.4%), but significant decreases in threonine (67.8%) and glutamine (56.7%), compared with the WT (Figure 7D).

### 3.7. Vital Role of 275 in Endocytosis, Septum Formation, and Mitochondrial Homeostasis

Considering the increased responses of Δ*275* to osmotic NaCl and sorbitol, as well as cell-wall-disturbing Congo red and SDS, combined with the most downregulated membrane transport and ABC transporters, endocytosis analysis was performed (Figure 8A). Hypha of both Δ*275* and the WT were stained with an endocytic tracer—the lipophilic styryl dye FM4-64, which was gradually internalized and stained endosomes and vacuoles after incubation. Δ*275* clearly displayed distinct endocytosis from the WT. The WT exhibited local discontinuities of strong endocytosis, while Δ*275* did not, suggesting that the lack of *275* resulted in dramatically decreased endocytosis. Interestingly, Δ*275* displayed swollen hyphae and OE-*275* had abnormal septa compared with the WT (Figure 8B). In particular, further septum dye assay suggested that septa with larger hollows were observed in the conidia of Δ*275* and OE-*275* compared to those in the WT, while the conidia of both Δ*275* and OE-*275* displayed thinner septa than the WT (Figure 8C). Transmission electron microscopy (TEM) analysis displayed that the numbers of mitochondria and vacuoles in Δ*275* were dramatically higher than those in the WT (Figure 8D). Moreover, mitochondrial hypertrophy, hyperplasia, and vacuolization were also observed in ∆*275* (Figure 8D). In addition, ∆*275* displayed amplified rough endoplasmic reticulum, concentric lamellar bodies, and lacked large vacuoles compared to the WT. Importantly, the WT strain consisted of many black tight bodies, whereas ∆*275* had few (Figure 8D). In order to evaluate the effects of varied vacuoles on the iron levels in the fungus, the levels of Fe^2+^, Fe^3+^, and total iron were also evaluated. However, no significant differences in the levels of Fe^2+^, Fe^3+^, and total iron were observed in ∆*275* compared to the WT. Interestingly, OE-*275* displayed increased levels of Fe^2+^, Fe^3+^, and total iron compared to the WT.

### 3.8. Docking of 275 with the PKS-PTS Hybrid Metabolites

To investigate the possible binding ability of the precursors, intermediates, and their derivatives from the PKS-PTS hybrid biosynthetic pathway in *A. oligospora* with the 275 protein, molecular docking analysis was performed. Interestingly, all of the 14 tested compounds displayed binding free energy values ranging from −3.0 to −6.75 kcal/mol (Figure 9). A previous study indicated that binding free energy less than −2 kcal/mol denotes good binding activity between ligands and receptors [31]. It is worth noting that the addition of the farnesyl moiety to toluquinol greatly improved the binding affinity to 275. Moreover, the PKS-PTS hybrid metabolites with the epoxy ring displayed the highest binding affinity (Figure 9 and Figure 10). Molecular visualization showed that the formation of hydrogen bonds between compounds and amino acid residues within the 275 protein can strengthen the interactions (Figure 10).

## 4. Discussion

Here, we found that the rare gene *275*, embedded in between the two biosynthetic genes *276* and *274* in the gene cluster *215g* (Figure 1A), was not involved in the biosynthesis of SEC metabolites in the nematode-trapping fungus. Notably, the gene *276* was responsible for the biosynthesis of the first hybrid precursor farnesyl–toluquinol by transferring a farnesyl unit to toluquinol (Figure 9). The gene *274* is a dehydrogenase that is involved in the oxidation modification of SECs [14]. A previous study suggested that toluquinol could be easily dehydrogenated to a more active toluquinone for the methylation, along with a pair of protons and electrons [14]. Interestingly, we also noted that the PKS-PTS hybrid gene clusters *Yan* in *Aspergillus* sp. and *Mac* in *Penicillium* sp., without the analogues of gene *275*, displayed distinct gene organizations from *215g* in *A. oligospora*. Interestingly, the analogues of *gene* 276 in the *Yan* and *Mac* clusters were just between the PKS gene and the decarboxylase gene that were involved in the biosynthesis of the first and second precursors—6-MSA and *m*-cresol, respectively—for the SECs. Importantly, all of the oxidoreductase genes including the analogues of gene *274* were arranged on the other side of the PKS gene, suggesting that gene *276* should be apart from *274* or other oxidoreductase genes, and that gene *275* might function to separate gene *276* from gene *274* in the *215g* cluster. Solid evidence from the varied resistances to H_2_O_2_ and GO enrichment analysis indicated that oxidoreductase activity and antioxidant cysteine-type endopeptidase activity were most regulated in Δ*275* and OE-275, respectively. In addition, the gene *276* was the most abundant upregulated gene in Δ*275*. In fact, all of the genes involved in the PKS-PTS hybrid pathway (273–283) were dramatically upregulated in Δ*275*. In particular, two oxidation genes—*279* and *277*, which were both involved in the formation of the key epoxy ring in the first PKS-PTS hybrid precursor—were the most upregulated. The metabolic profiles confirmed the dramatically increased SEC contents. These results suggested that the gene *275* played a key role in regulating the biosynthesis of SEC metabolites.

Bioinformatics analysis indicated that no homologues with high identity to 275 could be found in the known proteins. Detailed phylogenetic trees suggested that 275 shared a very low identity to the Rai1, cortactin, and Vit1 proteins. From the chemical structural point of view, arthrobotrisins shared much similarity to retinol (ligands 1–3 in Figure 9 and Figure 10). Molecular docking analysis suggested strong binding affinity between the epoxy PKS-PTS hybrid metabolites with protein 275, suggesting that 275 might be a Rai1-like protein. A previous study suggested that Rai1 is a transcription factor associated with Smith–Magenis syndrome, where deletion of Rai1 disrupted light-activated daytime melatonin suppression, with consequently high diurnal levels and low nocturnal levels of melatonin, and causing abnormal circadian signaling with genetic disorders [32]. Interestingly, the period circadian protein was by far the most abundant downregulated gene in the fungus with the disruption of 275. A recent study suggested that calcium signaling is pivotal to the circadian clockwork in the suprachiasmatic nucleus—particularly in rhythmic entrainment to environmental light–dark cycles [21]. We also found that the gene encoding neuromodulin—which is reportedly involved in Ca^2+^/CaM binding—was also among the top five downregulated genes in Δ*275* vs. the WT. All of these results suggest that *275* might be a *Rai1*-like gene.

Interestingly, transcriptional analysis also revealed that the key gene encoding Arp2/3, which was recruited by cortactin to play an important role in the cytoskeleton and endocytosis, was also the most downregulated. Further analysis revealed that the fungal mycelia, the septa of the fungal conidia, and endocytosis became abnormal in both mutants (Δ*275* and OE-275), together with distinct aerial morphology on the cell-wall-disturbing agent SDS, consistent with the downregulated gene *Arp2/3*. All of these results suggest that *275* might also be a *cortactin*-like gene.

A previous study suggested that Vit1 is a H^+^-coupled antiporter for metal ions, which provides the vacuolar H^+^-ATPase with an important function in metal storage [25]. A recent study reported that loss of vacuolar acidification impaired vacuolar cysteine storage, causing oxidative damage to mitochondrial iron–sulfur cluster biosynthesis components and activities, such as aconitase [33]. As a consequence, cells developed an age-related mitochondrial dysfunction characterized by the activation of the iron-deficiency and DNA-damage responses. Notably, a putative L-fucose-proton symporter gene was most downregulated in Δ*275* vs. the WT. Moreover, the mutants displayed distinct responses to NaCl and sorbitol, together with fast aging of the aerial mycelia of OE-*275* compared to the WT. Importantly, exceptional TCA cycles, mitochondrial dysfunctions, and unusual vacuoles were also observed in Δ*275*; for example, cis-aconitate, D-isocitrate, and α-ketoglutarate could be not detected in Δ*275*. These results indicate that *275* might be a *Vit1*-like gene.

In conclusion, our findings reveal that the gene *275* might function as a separator and as an integrated gene that might perform multiple functions related to three distinct proteins—Rail, cortactin, and Vit1—in the nematode-trapping fungus (Figure 11). Our findings provide insight into the intriguing organization of a rare gene that is likely derived from three genes, which might help to develop effective biological agents for combating nematode pests.

## Figures and Tables

**Figure 1 jof-08-01261-f001:**
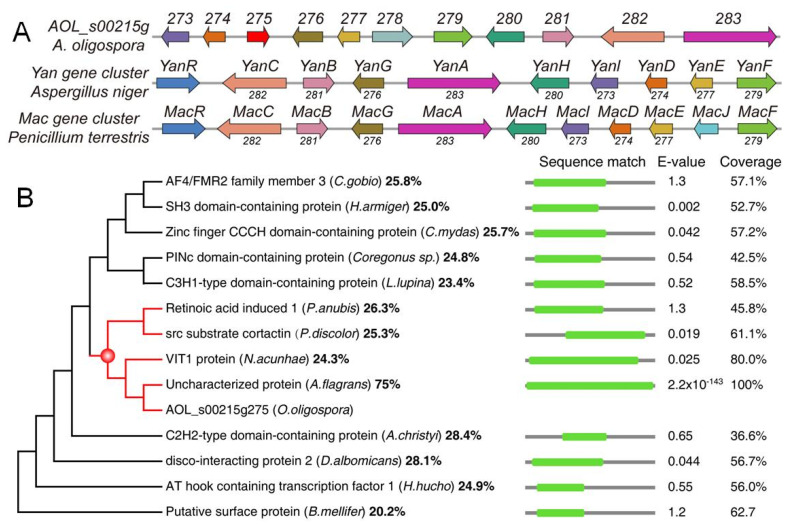
Organization of the genes in the *215g*, *Yan*, and *Mac* gene clusters for the PKS-PTS hybrid metabolites (**A**), and phylogenetic analysis of 13 sequences with high coverage and similarity to 275 (**B**).

**Figure 2 jof-08-01261-f002:**
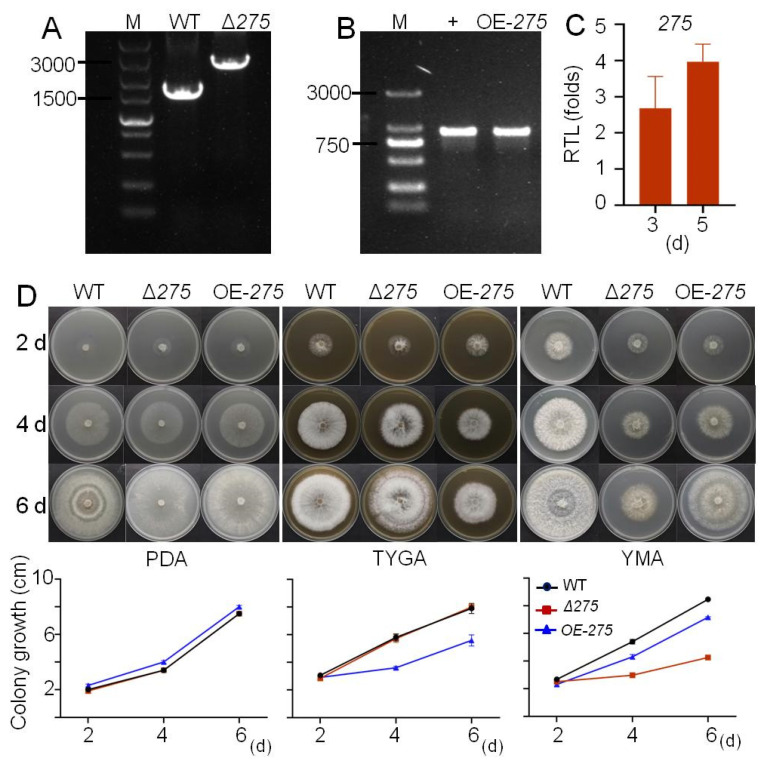
Confirmation of the mutant Δ*275* with the disruption of *275* and the mutant OE-*275* with the strong promoter replaced, and comparison of the colony growth of these two mutants with the WT on three media: (**A**,**B**) PCR analysis confirmed the mutants Δ*275* (**A**) and OE-*275* (**B**). (**C**) The relative transcriptional level (RTL) of *275* in the mutant OE-*275* at 3 and 5 days compared to that in the WT. (**D**) Comparison of the colony growth of these two mutants with the WT on three media (PDA, TYGA, and YMA).

**Figure 3 jof-08-01261-f003:**
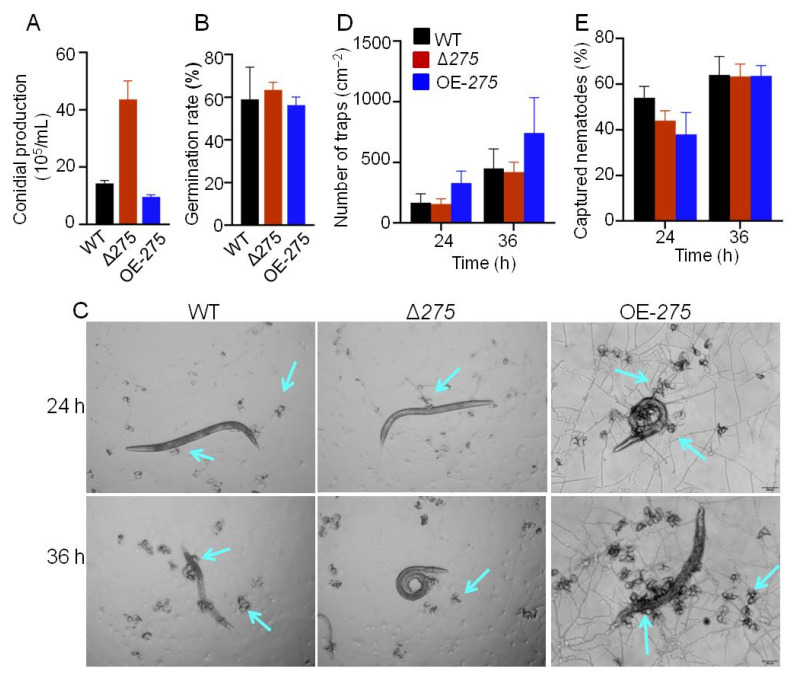
The effects of *275* on conidial production, germination, trap formation, and nematicidal activity: (**A**–**D**) Quantitative analysis of conidial production (**A**), germination rate (**B**), trap formation (**C**), and nematicidal activity (**D**), comparing the mutants Δ*275* and OE-*275* with the WT. (**E**) Comparison of trap formation nematicidal activity between the mutants Δ*275* and OE-*275* and the WT at 24 and 36 h. The arrows indicate the traps.

**Figure 4 jof-08-01261-f004:**
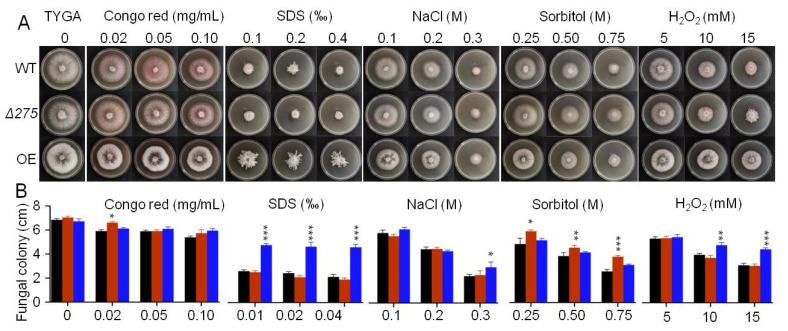
The effects of *275* on the fungal resistances to chemical stressors on TG plates: (**A**,**B**) Comparison of the colony growth between the two mutants (Δ*275* and OE-*275*) and the WT under treatment with Congo red, SDS, NaCl, sorbitol, and the oxidant H_2_O_2_ (*: *p* < 0.05; **: *p* < 0.01; ***: *p* < 0.001).

**Figure 5 jof-08-01261-f005:**
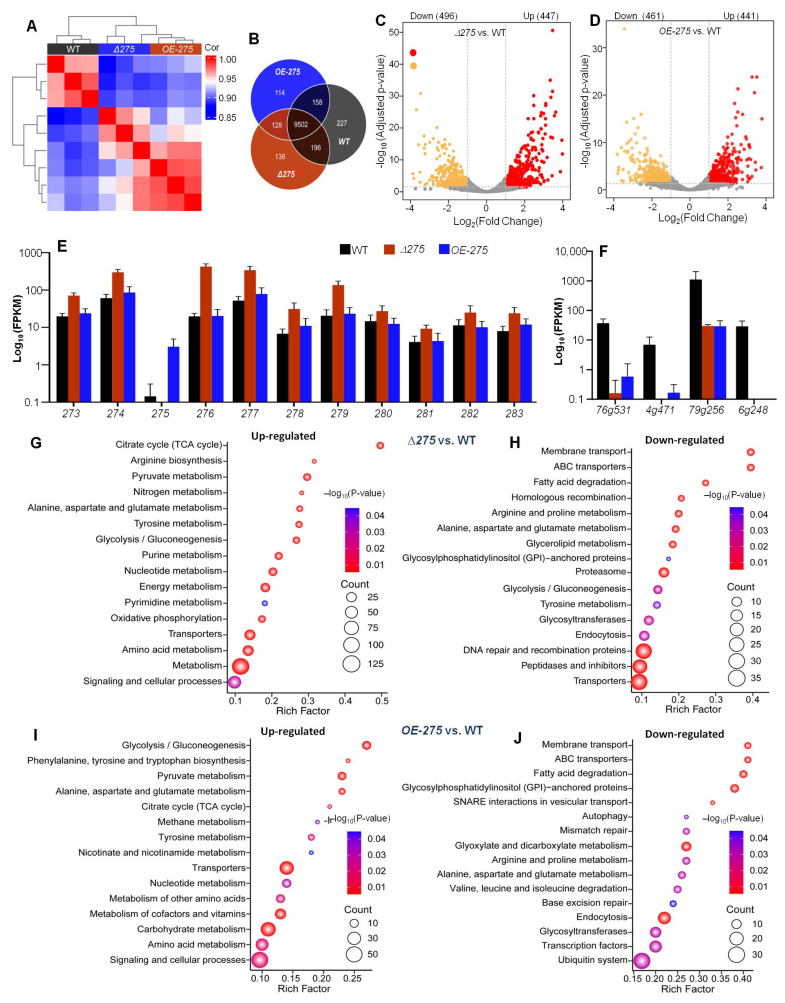
Transcriptomic analysis of Δ*275*, OE-*275*, and the WT: (**A**) Plot analysis of Pearson’s correlation (*r*) among the mutants Δ*275* and OE-*275* and the WT. (**B**) Venn analysis for the numbers of significantly differentially regulated genes in Δ*275* and OE-*275*, compared to the WT. (**C**,**D**) Volcano plot analysis of upregulated and downregulated genes in Δ*275* and OE-*275* vs. the WT, respectively. (**E**) Comparison of the transcriptional levels of the genes in the PKS-PTS hybrid pathway in Δ*275* and OE-*275*, compared to the WT. (**F**) Comparison of the transcriptional levels of the key downregulated genes in Δ*275* and OE-*275* compared to the WT. (**G**,**H**) KEGG analysis categorizing all of the significantly upregulated and downregulated genes in OE-*275* compared to the WT (relative abundance, log2 scale, *p* < 0.05). (**I**,**J**) KEGG analysis categorizing all of the significantly upregulated and downregulated genes in OE-275 compared to the WT (relative abundance, log2 scale, *p* < 0.05).

**Figure 6 jof-08-01261-f006:**
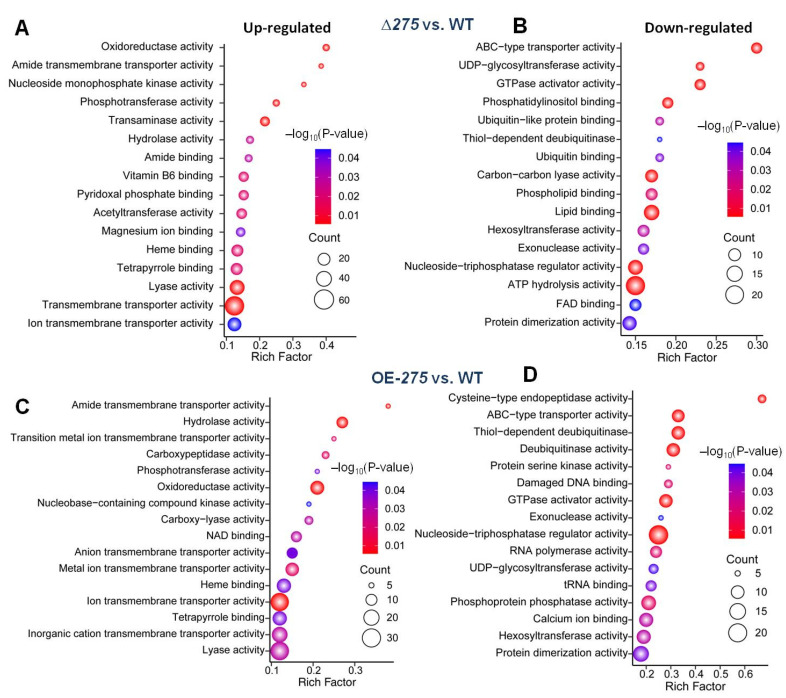
GO analysis of the significantly differentially regulated genes in Δ*275* and OE-275, compared to the WT: (**A**,**B**) GO analysis of the significantly upregulated genes (**A**) and downregulated genes (**B**) in Δ*275* vs. the WT. (**C**,**D**) GO analysis of the significantly upregulated genes (**C**) and downregulated genes (**D**) in OE*-275* vs. the WT.

**Figure 7 jof-08-01261-f007:**
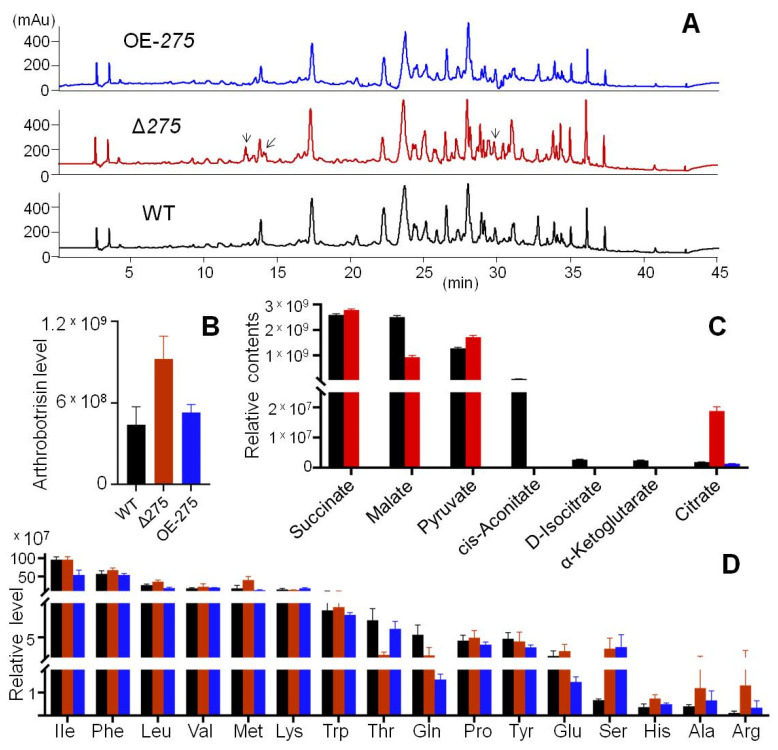
The effects of the gene *275* on the arthrobotrisin production, TCA cycle, and major amino acids: (**A**) HPLC profiles of the extracts of the liquid cultures of Δ*275*, *OE-275*, and the WT. The arrows indicate the extra peaks in Δ*275*. (**B**–**D**) Comparison of the levels of arthrobotrisin (**B**), major metabolites from the TCA cycle (**C**), and major amino acids (**D**) among Δ*275*, *OE-275*, and the WT.

**Figure 8 jof-08-01261-f008:**
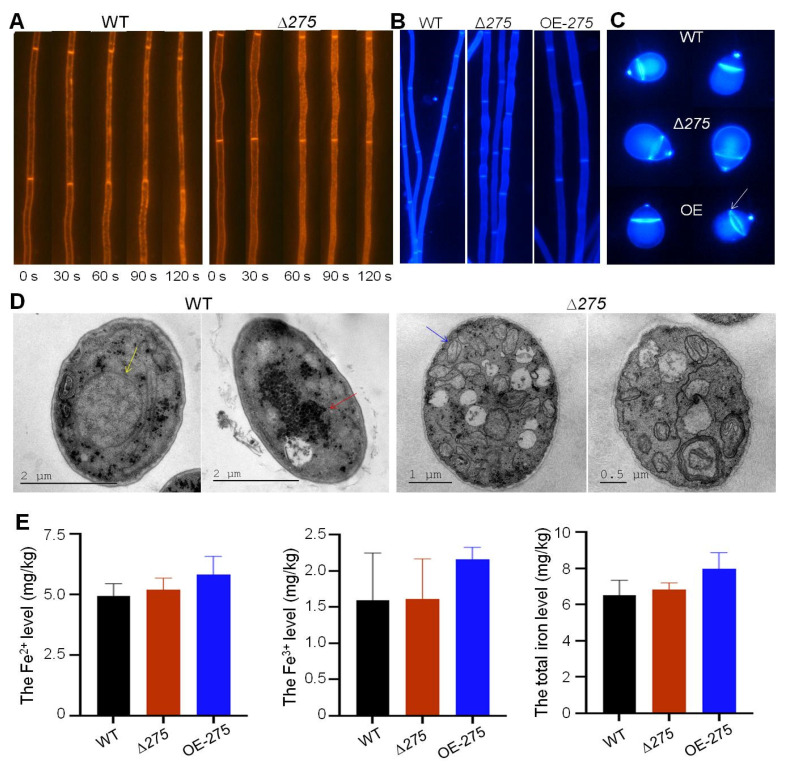
The effect of *275* on the fungal endocytosis, septum formation, infrastructure, and iron levels: (**A**) Comparison of the fungal endocytosis between Δ*275* and the WT within 120 s using the lipophilic styryl dye FM-464. (**B**) Comparison of the fungal mycelial morphology and septum formation between the mutants Δ*275* and OE-*275* and the WT using CFW staining. (**C**) Comparison of the conidial septum formation between the mutants Δ*275* and OE-*275* and the WT. White arrow: septum. (**D**) TEM analysis of the infrastructures between the mutant Δ*275* and the WT. Yellow arrow: vacuole; Blue arrow: mitochondria; red arrow: black particles. (**E**) Comparison of the Fe^2+^, Fe^3+^, and total iron levels between the mutants Δ*275* and OE-*275* and the WT.

**Figure 9 jof-08-01261-f009:**
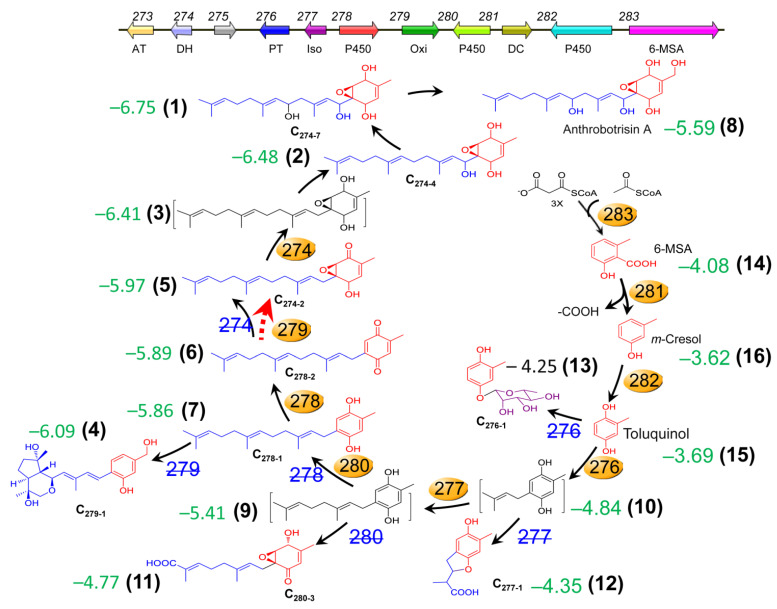
The hybrid metabolites derived from the PKS-PTS hybrid biosynthetic pathway with binding free energy values (referring to the binding affinity of SEC metabolites with 275) predicted by molecular docking analysis. All of the metabolites displayed binding free energy values below −3 kcal/mol.

**Figure 10 jof-08-01261-f010:**
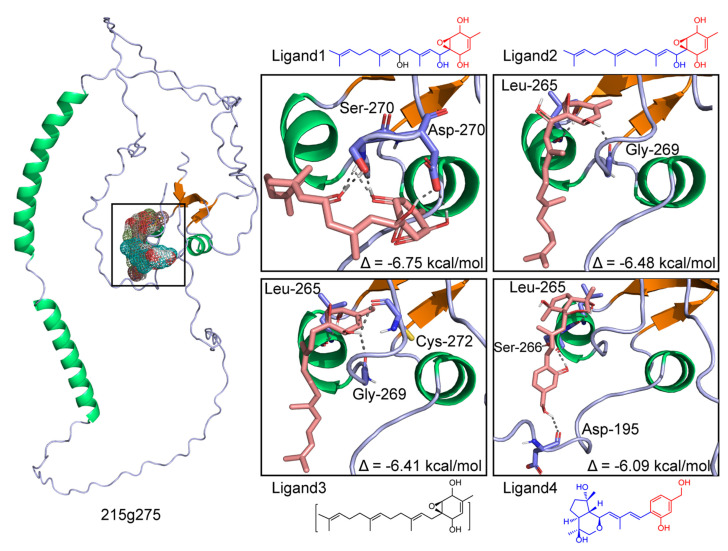
Molecular docking analysis of the strong binding affinity of the SEC metabolites derived from the PKS-PTS hybrid biosynthetic pathway with 275 (binding free energy values below −6 kcal/mol).

**Figure 11 jof-08-01261-f011:**
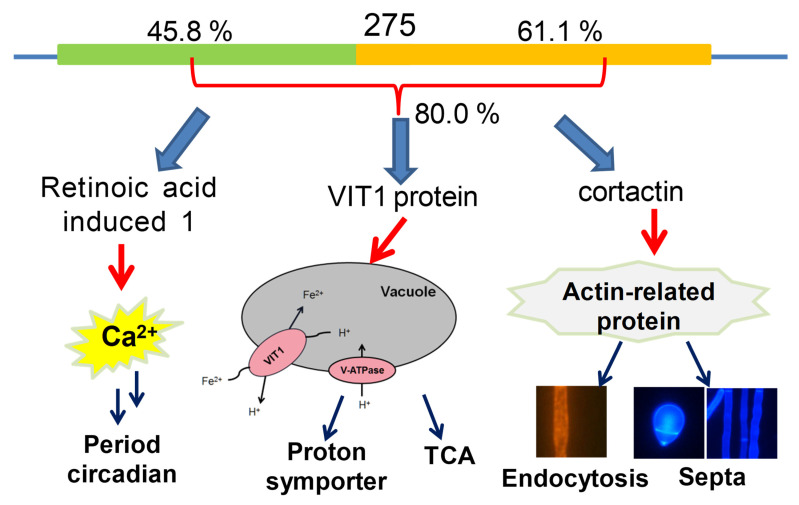
The proposed functions of 275 in the nematode-trapping fungus *A. oligospora*.

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
