# Peer review of "The Multifaceted Gene 275 Embedded in the PKS-PTS Gene Cluster Was Involved in the Regulation of Arthrobotrisin Biosynthesis, TCA Cycle, and Septa Formation in Nematode-Trapping Fungus Arthrobotrys oligospora"

_jof, 2022, doi:10.3390/jof8121261_

Round 1

Reviewer 1 Report

Title: Manuscript title is clear

Key words: Arthrobotrys oligospora please omit the already mentioned in the title

Abstract: The abstract is well written

The methods, results, and discussion sections were all-sufficient and clear. The writing style is good and the command of the English language for the most part is excellent. Some editorial changes are needed

The tables and figs complement the manuscript. No additional tables and figs are needed. 

Author Response

Dear Reviewer,

Thank you very much for taking time out of your busy schedule to read our article. Thank for providing the constructive modification comments and suggestions, which have greatly helped us to further improve this manuscript. We carefully thought and analyzed your valuable opinions. On this basis, we have made the appropriate revisions and responses to your comments and suggestions, as noted below.

We have checked out the manuscript carefully, and polished some English language and styles scattered in the manuscript.

With these revisions to our manuscript, we trust that we have improved the submission. We sincerely hope it now meets all the requirements of the journal.

Thanks for your great efforts.

Best wishes!

Xuemei NIU

Reviewer 2 Report

The article is above average and very well written and deserves to be published in the form that it is.

Author Response

Dear Reviewer,

Thank you very much for taking time out of your busy schedule to read our article. Thank for your valuable comments on this manuscript. 

Thanks for your great efforts.

Best wishes!

Xuemei NIU

Reviewer 3 Report

I read carefully a very interesting and comprehensive research work entitled The Multifaceted Gene 275 Embedded in the PKS-PTS Gene Cluster was Involved in the Regulation of Nematicidal Activity, Arthrobotrisin Biosynthesis, TCA Cycle, and Septa Formation of Nematode-Trapping Fungus Arthrobotrys oligospora.

The concept of the manuscript is novel and perfectly suitable to publish in JOF.

Some minor corrections are required:

Title should be more precise the present form is very lengthy

Introduction should be modified and add more details and importance of the study with the recent review of literature.

If possible add graphical abstract which can represent whole research work

Author Response

Dear Reviewer,

Thank you very much for taking time out of your busy schedule to read our article. Thank for providing the constructive modification comments and suggestions, which have greatly helped us to further improve this manuscript. We carefully thought and analyzed your valuable opinions. On this basis, we have made the appropriate revisions and responses to your comments and suggestions, as noted below.

  1. Title should be more precise the present form is very lengthy

Response: “Nematicidal activity” in the Title has been deleted. Title has been shorten to “The Multifaceted Gene 275 Embedded in the PKS-PTS Gene Cluster was Involved in the Regulation of Arthrobotrisin Biosynthesis, TCA Cycle, and Septa Formation of Nematode-Trapping Fungus Arthrobotrys oligospora”

  1. Introduction should be modified and add more details and importance of the study with the recent review of literature.

Response: To improve the introduction and better highlight the importance of the study. We added the following in the manuscript: It is not unusual that there are ‘other’ incongruous genes embedded in a biosynthetic gene cluster that was responsible for the biosynthesis of secondary metabolites. Previous studies suggested that such ‘other’ genes appeared to be unnecessary for the formation of secondary metabolites but display important roles in the fungal self-protection mechanisms, including duplication of the metabolite target, detoxification of the metabolites and efflux of the metabolites [1]. However, limited attention has been paid to the functions of such genes in new biosynthetic gene clusters.

  1. If possible add graphical abstract which can represent whole research work.

Response: A graphical abstract which can represent whole research work has been provided in the revised manuscript as Figure 11.

Figure 11. The proposed functions of 275 in nematode-trapping fungus A. oligospora.

We have checked out the manuscript carefully, and polished some English language and styles scattered in the manuscript.

With these revisions to our manuscript, we trust that we have improved the submission. We sincerely hope it now meets all the requirements of the journal.

Thanks for your great efforts.

Best wishes!

Xuemei NIU

Reviewer 4 Report

I do like this manuscript, and I consider highly relevant for this special issue.

I consider that this manuscript has to be accepted after a minor linguistic review.

Author Response

(The authors gave the same response as above.)
